# A Comprehensive Survey on Wi-Fi Sensing for Human Identity Recognition

Pengsong Duan [ID], Xianguang Diao, Yangjie Cao, Dalong Zhang, Bo Zhang *[ID] and Jinsheng Kong

School of Cyber Science and Engineering, Zhengzhou University, Zhengzhou 450002, China; duanps@zzu.edu.cn (P.D.); diaoxianguang@gs.zzu.edu.cn (X.D.); caoyj@zzu.edu.cn (Y.C.); iedlzhang@zzu.edu.cn (D.Z.); jskong@zzu.edu.cn (J.K.)
* Correspondence: zhangbo2050@zzu.edu.cn

**Abstract:** In recent years, Wi-Fi sensing technology has become an emerging research direction of human–computer interaction due to its advantages of low cost, contactless, illumination insensitivity, and privacy preservation. At present, Wi-Fi sensing research has been expanded from target location to action recognition and identity recognition, among others. This paper summarizes and analyzes the research of Wi-Fi sensing technology in human identity recognition. Firstly, we overview the history of Wi-Fi sensing technology, compare it with traditional identity-recognition technologies and other wireless sensing technologies, and highlight its advantages for identity recognition. Secondly, we introduce the steps of the Wi-Fi sensing process in detail, including data acquisition, data pre-processing, feature extraction, and identity classification. After that, we review state-of-the-art approaches using Wi-Fi sensing for single- and multi-target identity recognition. In particular, three kinds of approaches (pattern-based, model-based, and deep learning-based) for single-target identity recognition and two kinds of approaches (direct recognition and separated recognition) for multi-target identity recognition are introduced and analyzed. Finally, future research directions are discussed, which include transfer learning, improved multi-target recognition, and unified dataset construction.

**Keywords:** Wi-Fi sensing; human identity recognition; received signal strength; channel state information; machine learning





## 1. Introduction

Identity is a critical issue in information security and an important element of pervasive computing. In the era of human-centered computing, human–computer interaction has been changing from "explicit interaction" to "implicit interaction", "explicit interaction" refers to the direct and conscious communication between users and computer systems, where users provide clear and intentional input. In contrast, "implicit interaction" represents a more indirect and natural form of communication, where technology can sense and respond to users' actions or behaviors without requiring explicit commands. Efficient identity recognition has always been the focus of academia and industry in human–computer interaction. Traditional identity-recognition methods, such as memorized passwords and ID cards, exhibit shortcomings such as being forgotten, inconvenienced in carriage, and being stolen. Over the past few decades, people have been developing new technologies to achieve more effective identity recognition. New identity-recognition technologies based on computer vision (CV), infrared, specialized sensors, and biological detection have been gradually studied and applied to daily life. The sensing principles of these applied methods are introduced in Section 2.

However, these new technologies exhibit certain shortcomings of their own. Specialized sensors and biological detection-based methods are contact-based, leading to unsatisfactory human experiences. With the development of contactless-based technologies, face

recognition, iris recognition, and other more efficient identity-recognition methods have started to emerge. However, the vision-based identity-recognition methods demonstrate some disadvantages in practical application, such as illumination/obstacle sensitivity, user privacy leakage, and limited recognition range. The infrared-based methods require a specialized device, which is financially unfriendly. To solve these problems, researchers have been working on more convenient, secure, and effective identity-recognition technologies.

In recent years, Wi-Fi sensing technology has become a major direction of sensing research with its advantages of cost-effectiveness, contactless, illumination insensitivity, better privacy preservation, and so on. The propagation paths of Wi-Fi signals can be categorized into line-of-sight (LOS) paths and non-line-of-sight (NLOS) paths. LOS paths refer to direct signal transmission from the transmitter to the receiver without obstruction, while NLOS paths involve signal reflections, refractions, and other pathways caused by obstacles during propagation [1]. In 2000, Bahl et al. [2] proposed Radar, becoming the first to utilize Wi-Fi-received signal strength (RSS) information for indoor positioning. Since then, the Wi-Fi RSS has been widely studied for human detection and recognition [3–5]. Because RSS depends on the transmitting power and is easily affected by multipath effects, it is more suitable for coarse-grained sensing tasks instead of fine-grained ones. Therefore, a better Wi-Fi sensing data carrier with high precision and robustness is in high demand. In 2011, Halperin et al. [6] released the channel state information (CSI) Tool for extracting CSI from commercial Wi-Fi devices, greatly facilitating the obtainment of CSI. It also shifts the research focus to fine-grained sensing tasks. Subsequently, there have been extensive efforts on human identity recognition based on Wi-Fi CSI, with emerging applications such as gait recognition [7–24], gesture recognition [25–30], sleep monitoring [31–33], and fall detection [34,35].

In addition to Wi-Fi sensing technology, there are also other identity-recognition methods based on radio frequency technologies, such as RFnet [36] using RFID technology, BLE-DoorGuard [37] using Bluetooth technology, and ARDEA [38] using ZigBee technology. Wi-Fi signal, enjoying high signal penetration ability, wide coverage, and low deployment cost, has recently attracted extensive attention from both academia and industry, for realizing precise and reliable human identity recognition based on a large number of ready-to-use commercial Wi-Fi facilities being actively studied.

As shown in Table 1, compared with traditional identity-recognition technologies, Wi-Fi sensing technology enjoys many advantages.

**Table 1.** Comparison of various identity-recognition technologies.

| Technology | Applications | Sensing Range | Cost | Accuracy |
|---|---|---|---|---|
| Computer Vision | Face, fingerprint, iris, gait, etc. | LOS | High | Middle |
| Infrared | Near-infrared, mid-infrared, far-infrared, etc. | NLOS | High | High |
| Specialized Sensor | Speed sensor, ground sensor, etc. | Sight Distance | High | High |
| Biological detection | DNA, body odor, etc. | Contact | High | High |
| Wi-Fi | Gait, gestures | NLOS | Low | Middle |
| RFID | RFID tag | NLOS | High | High |
| Bluetooth | Gait | LOS | Low | Low |
| ZigBee | Gait | LOS | High | High |

There are some review papers for Wi-Fi sensing technology, such as articles [39–42]. The main focus of these articles is on either activity recognition or human detection and sensing methods for single-target scenarios. As an important part of the Wi-Fi sensing fields, a review paper that focuses on human identity recognition is very rare and necessary. How to achieve high-precision and highly reliable human identity recognition based on existing Wi-Fi utilities has become a research topic that scholars are very concerned about. To further enhance the attention and understanding of Wi-Fi perception issues and promote the continuous development of Wi-Fi perception technology, this article, based on surveying a large number of related works, provides a detailed introduction and analysis

of the existing human identity-recognition technology based on Wi-Fi signals, including basic processes, main methods, and future developments.

To ease reading, we summarized the major abbreviations of technical terms used in this paper in Table 2.

**Table 2.** Major abbreviations used in this paper.

| Acronym | Full Name |
| --- | --- |
| ECG | Electrocardiogram |
| HDID | Humanitarian DNA Identification Database |
| RSS | Received Signal Strength |
| CSI | Channel State Information |
| CFR | Channel Frequency Response |
| OFDM | Orthogonal Frequency Division Multiplexing |
| KNN | K-nearest Neighbor |
| SVM | Support Vector Machine |
| LSTM | Long Short-Term Memory |
| CNN | Convolutional Neural Network |
| GRU | Gated Recurrent Unit |
| DWT | Discrete Wavelet Transform |
| PCA | Principal Component Analysis |
| DTW | Dynamic Time Wrapping |
| SAC | Sparse Approximation Classification |
| GMM | Gaussian Mixture Model |
| RBF | Radial Basis Kernel Function |
| HMM | Hidden Markov Model |
| STFT | Short Time Fourier Transform |
| IFFT | Inverse Fast Fourier Transform |
| SVDD | Support Vector Domain Description |
| DFS | Doppler Frequency Shift |
| RF | Random Forest |
| IDP | Iterative Dynamic Programming |
| MUSIC | Multiple Signal Classification |
| PF | Particle Filter |
| JPDAF | Joint Probability Data Association Filter |
| NIC | Network Interface Card |
| LOS | Line-of-Sight |

The remainder of this paper is organized as follows. Section 3 concentrates on the basic process of human identity recognition based on Wi-Fi signals, including data acquisition, data pre-processing, feature extraction, and identity recognition. Sections 4 and 5 review the state-of-the-art identity-recognition methods based on Wi-Fi sensing technology for both single-target and multi-target scenarios. Section 6 highlights future research directions. Section 7 concludes this paper.

## 2. Current Methods for Human Identity Recognition

At present, the main applied methods for human identity recognition in practice often adopt technologies that are based on CV, infrared, specialized sensors, and biological detection, respectively. Generally speaking, the sensing principles of these methods can be mainly categorized as follows.

### 2.1. Computer Vision

In most CV-based methods, the typical biometric image sequence of a human body is first collected through a camera device; then computer vision computing is adopted to extract and identify human feature information, such as face [43–45], fingerprint [46,47], iris [48,49], gait [50,51], etc. In 2019, Wei et al. [52] utilized deep learning techniques, employing spatial and temporal attention pooling networks to eliminate redundant information from videos and automatically determine a person's identity. The spatial attention

network was used to extract spatial feature maps, while the temporal attention network was employed to capture temporal information. Person identity recognition was achieved by computing the distance between features.

### 2.2. Infrared

The basic principle of infrared-based methods is to adopt a human body infrared thermal-imaging approach, fulfilling identity sensing in low-light scenarios [53–55]. Infrared thermal imaging can meticulously observe musculoskeletal and blood vessels as well as sense the skin temperature of a human body [56]. In addition, due to the multipath effects, the recognition accuracy of infrared technology is higher than that of Wi-Fi sensing. Although infrared-based identity-recognition approaches are both illumination insensitive and contactless, the limited sensing range and the requirement of expensive dedicated equipment restrain them from wide deployment.

### 2.3. Specialized Sensors

With specialized sensors, human identity recognition can be achieved by collecting and analyzing a wide range of sensed data using sensors such as accelerometers [57,58] and ground sensors [59–61] attached to the human body. In 2019, Shunmugam et al. [62] proposed a human recognition system based on a combination of Kinect sensors, in which an IRdepth sensor, an RGB camera, and a microphone are used for bone recognition, facial recognition, and voice recognition, respectively. Although this system can realize accurate identity recognition, it is difficult to be widely implemented due to its high cost, inconvenient installation, and low portability.

### 2.4. Biometric

Biological characteristics are important signs of the human body. Through the detection and matching of human biological attributes such as DNA [63,64] and body odor [65,66], accurate human identity recognition can be realized. In 2020, Budowle et al. [67] created an HDID for the identity recognition of human remains and missing persons. The main disadvantages of bio-based approaches include expensive detection equipment and long chemical examination delays.

## 3. Main Process for Human Identity Recognition Using Wi-Fi Signal

Most Wi-Fi-based identity-recognition approaches are dedicated to action recognition, namely sensing identity, by analyzing the signal perturbation characteristics of human actions (such as gait, gestures, etc.). According to the number of targets, the corresponding research can be categorized into single-target identity recognition and multi-target identity recognition. As shown in Figure 1, the process of Wi-Fi sensing-based human identity recognition normally includes four steps: data acquisition, data pre-processing, feature extraction, and classification. Firstly, signal data reflecting human motion characteristics are obtained from a Wi-Fi signal receiver. Secondly, collected data are pre-processed to reduce signal noise. Thirdly, certain algorithms are applied to separate the effective fragments containing gait or gesture information, followed by performing feature extraction. Finally, a trained feature classifier is used for identity classification.

### 3.1. Data Acquisition

Signal data acquisition is the first step for Wi-Fi sensing. Effective and accurate collection of the signal containing human action information directly influences the identity-recognition result. A wireless signal acquisition device generally consists of a few transmitters and a few receivers. The transmitter can be a commercial Wi-Fi device in general, while the receiver is usually a computer equipped with a wireless card, i.e., Inter 5300 NIC [68]. There are mainly two kinds of human motion characteristic signals in Wi-Fi sensing, namely RSS and CSI.

1. RSS refers to the received signal strength indicator at the MAC layer, and RSS is usually used to decide the necessity of transmission power increment. The RSS information has been extensively explored for human identity recognition in [3–5].
2. CSI refers to channel state information, which estimates the channel gain in OFDM technology and is also the result of sampling channel frequency response (CFR) [69]. In OFDM systems, CSI is represented at the subcarrier level and includes the amplitude and phase information. CSI can be regarded as RSS information modulated by OFDM technology and contains richer information. Assuming that the numbers of antennas at the transmitter and the receiver are $N_t$ and $N_r$, respectively, and the number of subcarriers is m, the receiver can parse out $N_t \times N_r \times m \times T$ CSI values in time slot $T$.

Compared to RSS, CSI has the following advantages:

- CSI is more sensitive to changes in the surrounding environment and provides highly accurate information, thus facilitating fine-grained sensing for more potential applications.
- CSI contains both amplitude and phase information in each subcarrier.
- Due to the use of OFDM technology, CSI is less susceptible to multipath effects than RSS.

In conclusion, CSI has become the mainstream sensing carrier in current Wi-Fi sensing research due to its better detail sensing ability. By using amplitude and phase information in CSI, an increasing number of human identity-recognition approaches have emerged, such as WiWho [7] and FreeSense [12].

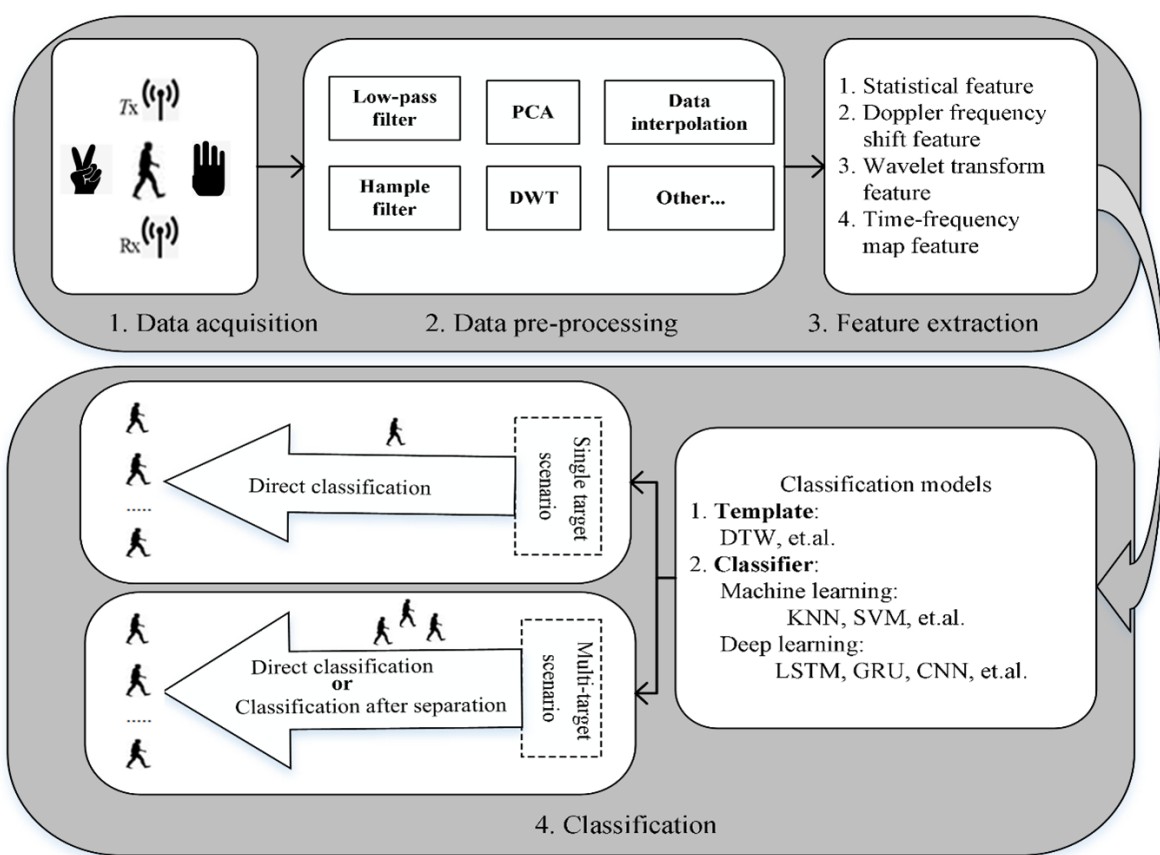

**Figure 1.** The basic process of Wi-Fi Sensing for Human Identity Recognition.

### 3.2. Data Pre-Processing

It is well known that the measurement of CSI or RSS includes useful signals, disordered noise, and some outliers. These mixed data mainly come from the effects of the surrounding environment, signal interference, and human disturbance to the signal. Therefore, the method of data pre-processing is very important as it can remove outliers, filter noise,

calibrate the phase, and retain valuable data. Currently, various signal pre-processing techniques can be used to achieve denoising, data interpolation, and phase cleaning. Generally speaking, the premise of identification is to obtain accurate data. Therefore, before extracting features, the data needs to be pre-processed. Specifically, to reduce invalid data and improve identification accuracy, it is necessary to remove the multipath effect and the resulting noise. There are several typical denoising methods, such as low-pass filter, principal component analysis (PCA), data interpolation, Hample filter, and Discrete Wavelet Transform (DWT).

### 3.2.1. Low-Pass Filter

Low-pass filter allows signals below the cutoff frequency to pass through while filtering out high-frequency signals, to retain only low-frequency characteristic information that reflects human interference on Wi-Fi signals. Commonly used low-pass filters include the Butterworth filter and Gaussian filter. An ideal low-pass filter often has very sharp transitions and produces ringing [70], while a Gaussian low-pass filter is smoother and does not produce ringing. The smoothness of the Butterworth low-pass filter is between the ideal low-pass filter and the Gaussian low-pass filter, thus, it is a commonly used filter in wireless sensing. Although the low-pass filter has good performance, there are still some noises that cannot be effectively removed. These noises usually come from the transmitter and receiver or environmental changes.

### 3.2.2. Principal Component Analysis

PCA is a data processing method that attempts to represent data with partial features of a matrix, reducing data dimensions and enhancing processing efficiency. Specifically, PCA can select some eigenvectors to construct a new matrix representing the original matrix by computing its eigenvalues. In CSI-based identity recognition, PCA is mainly used to eliminate noises and data redundancy in the signal processing step. PCA can be implemented based on either the eigenvalue decomposition or SVD decomposition of the covariance matrix. For example, WiGrus [71] uses the PCA method to pre-process the received data for gesture recognition.

### 3.2.3. Data Interpolation

In practical applications, although the data transmission rate of the sender is constant, the receiver usually cannot receive data reliably, due to data loss or transmission delay. Therefore, effective measures must be taken to solve this problem. Data interpolation is an effective way to solve this problem. Specifically, for each missing data, a synthesized sample computed from neighboring data is used to fill the slot, thereby constructing a uniform CSI sequence, eliminating data clustering and ambiguity, and improving feature extraction accuracy. For example, MSM [72] uses an interpolation method to pre-process the received data and implement robot indoor positioning.

### 3.2.4. Hample Filter

At present, there are many outliers collected in CSI data due to sudden changes in equipment and environment. The Hample filter is an effective technique to remove outliers from data that are isolated from its neighbors. Specifically, it finds outliers and replaces them with data averaged by utilizing a moving average window, thus eliminating the negative effects of invalid data. In the case of CSI signals, outliers caused by the device and environment can be erased by using the Hample filters. For example, WiHACS [73] uses the method to pre-process the received data to realize the classification of human activities.

### 3.2.5. Discrete Wavelet Transform

DWT is a discretization of the scale and translation of the basic wavelet. It can be used in image processing because it can remove noise in the signal, and extract and retain some useful edge information. It can overcome the shortcomings of traditional Fourier-

transform-based image processing, including the effect of local abrupt changes and loss of image edge information when removing signal noise. For example, Wihi [74] uses the DWT method to pre-process the received data to realize human identity recognition.

### 3.3. Feature Extraction

Feature extraction is used to choose effective features from feature sets. Pre-processed wireless sensing data contain effective information that reflects disturbance features of human movements on the signals, which needs to be extracted for the identity classifier. For the Wi-Fi sensing-based human identity recognition, the typical features include the Statistical feature, Doppler frequency shift feature, Wavelet transform feature, and Time-frequency map feature [75], as shown in Table 3.

**Table 3.** Comparison of different features.

| Feature Name | Feature Extraction Principle | Feature Analysis |
|---|---|---|
| Statistical feature | Statistical analysis of collected data | Low complexity, clear data correlation, inadequate feature extraction |
| Doppler frequency shift feature | Measurement of Doppler frequency shift movement | Distinct differentiation |
| Wavelet transform feature | Analysis of signals at multiple frequency scales | Fine-grain and multi-scale feature extraction |
| Time-frequency map feature | Analysis of cross-domain characteristics | Rich signal feature information and intuitive display |

### 3.3.1. Statistical Feature

The statistical feature mainly includes maximum, minimum, mean, variance, mean square root, and frequency distribution statistics of signal in the time domain, and Fourier transform value, spectral probability, signal energy, spectral entropy, and frequency peak statistics of signals in the frequency domain. In the early stage of Wi-Fi sensing, statistical methods were often used to extract signal features. For example, in the works [10,76], difference threshold estimation and mean absolute deviation were used to extract signal features to make the contrast between them more obvious, as shown in Figure 2.

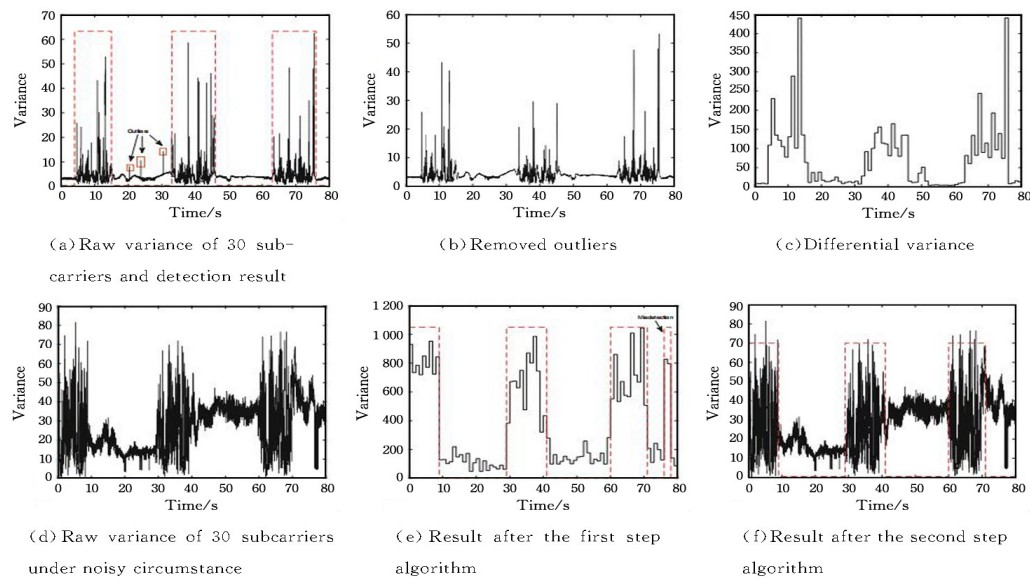

**Figure 2.** Statistical characteristics [10].

### 3.3.2. Doppler Shift Feature

Detecting identity information in an indoor environment requires humans to complete certain motion actions, which often cause Doppler shifts. For example, Pu et al. [76]

proposed WiSee, which performs gesture recognition by extracting Doppler frequency shift features.

### 3.3.3. Wavelet Transform Feature

Wavelet transform eases the signal analysis spanning over a wide frequency band, making it possible for fine-grain feature extraction. For example, Yunfang et al. [77] used wavelet transform to extract features of different frequency bands to obtain motion speeds corresponding to different body parts.

### 3.3.4. Wavelet Transform Feature

By using STFT and other algorithms, action signals can be transformed into time-frequency maps to obtain richer feature information for higher recognition accuracy. In the WifiU [8] system, the processed CSI signal is converted into the time-frequency domain, and then the waveform is converted into the time-frequency diagram through STFT transformation, as shown in Figure 3.

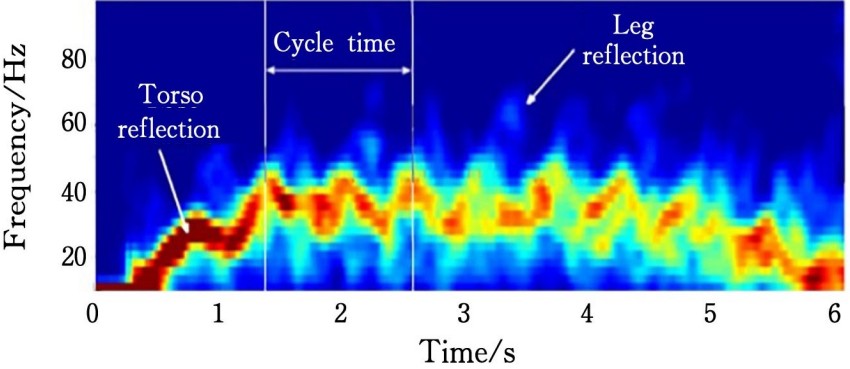

**Figure 3.** Time-frequency diagram [8].

### 3.4. Classification

After extracting features for each action and creating a database of motion templates, newly acquired sensing signals can be analyzed for identity recognition. The sensing methods of human identity can be divided into template recognition and classifier recognition. The template identity-recognition method usually adopts the DWT algorithm to directly calculate the similarity between newly acquired signals and templates and then finds out the closest class. In classifier recognition, supervised learning is often adopted, in which sensing data are collected and labeled, and then the classifier is trained to recognize the newly collected signals. The classifier methods include SVM, KNN, and CNN, among others, which can be divided into machine learning and deep learning, as shown in Table 4. In general, machine learning includes deep learning, but machine learning in this paper means traditional machine learning, which excludes deep learning.

The classification of identity information is usually the final stage of research, and verifying the performance of the designed model is also necessary. Both conventional machine learning and deep learning need to train classifiers in advance to form a mapping for later tests and recognition. In addition, classifiers are often needed to consider whether the application scenario is a two-class or multi-classification problem. Generally speaking, such as logistic regression and SVM are often used to solve binary classification problems, and the SoftMax function is often used for multi-classification problems. Combined with the actual scenario of Wi-Fi sensing, the identified applications are usually multi-classification problems.

**Table 4.** Commonly used feature classification algorithms.

| Category | Name | Principle | Advantages | Disadvantages | Applications |
|---|---|---|---|---|---|
| Machine Learning | DTW | Calculate the similarity between time series data | No pre-training requirement; fast match | More computationally intensive; more dependent on templates | FreeSense [12], Literature [28] |
| | KNN | the category of k samples that are most close to it | Intuitively simple; easy to implement | Performance degradation when data samples are not evenly distributed; high storage space usage | MAIS [10], FreeSense [12] |
| | SVM | Solve for separated hyperplane that correctly partitions a training dataset | High precision and good generalization ability | Inapplicability to Large Sample Data | WifiU [8], Wii [11], SiWi [26], WiGA [24], WiDIGR [18] |
| Deep Learning | LSTM | Model relationships through gating mechanisms | Suitable for processing time series data | Classifier training is time-consuming | CSIID [15], WiID [25], FingerPass [29] |
| | GRU | A simplified version of LSTM that uses hidden states | Fewer parameters than LSTM, and not easy to overfit | In the same data sets, LSTM performs better | WiID [17], Deep-WiID [19], WiHF [30] |
| | CNN | Extract multi-dimension features by convolution kernel | Strong feature extraction ability | Classifier training is time-consuming | WiNet [21], TransferSense [24] |

## 4. Single-Target Scenario

Most of the existing identity-recognition methods using Wi-Fi signals are designed for single-target scenarios, as this scenario is relatively simple without the need to consider signal separation. The multi-target scenario is a completely different story. In Wi-Fi environments with multiple targets, the received signals captured by the receiver include not only multipath signals from surrounding obstacles but also multipath signals generated by mutual reflection between human bodies. When individuals are nearby, the receiver may mistakenly perceive the signals from multiple people as a single unified signal, leading to failures in recognizing multiple individuals. Additionally, the separation of signals from multiple individuals poses a significant challenge. In multi-person scenarios, the system needs to accurately separate and identify signals generated by different individuals to ensure the precise identification of each person, adding complexity and difficulty to identity-recognition systems. Section 5 provides a detailed explanation of multi-person identity recognition. According to the target action granularity, the identity-recognition methods of single-target scenarios can be categorized into two types, gait-based and gesture-based. The existing studies have shown that gaits and gestures are important biological characteristics that define human identity. Gait recognition technology collects dynamic features of body parts, which are richer than static features. Due to differences in body shape and movement patterns, Wi-Fi signals can be interfered with by gait in a specific area, resulting in highly recognizable characteristic responses to the received signal [12]. Although gestures and gaits are the main biological characteristics of the human body, the same gesture is not as distinct among different populations as gait, making it difficult to identify. Utilizing gestures for identity recognition is built upon the foundation of gesture classification. Therefore, most research efforts in the field of gestures, no matter classification or identity recognition, are primarily focused on the recognition of different types of gestures. On the other hand, mere classification of gestures is inadequate in identification, as it fails to map each gesture (even from the same class) to each individual. Human identification, therefore, requires not only gesture classification to begin with, but also need to extract sufficiently fine-grained features from each gesture to complete such mapping. In recent years, with the continuous development of Wi-Fi sensing technology, there has been positive progress in gesture-based identity-recognition techniques. For example, WiID [17], SiWi [26], FingerPassf [29], WiHF [30].

According to the classification method, identity-recognition methods can be divided into three types: pattern-based, model-based, and deep learning-based. The pattern-based method, which usually requires more data to fit or train network parameters, utilizes pattern recognition methods, such as machine learning, to recognize human identity. Differently,

the model-based method exploits mathematical or physical models to describe the unique relationship between CSI data changes and human identity. Therefore, the key issue of the model-based method is how to extract effective signal features to express human identity and build a model to describe the signal changes caused by human activities. The deep learning-based method requires a large amount of experimental data to train a neural network because it usually contains millions of parameters. Furthermore, the deep learning-based method can automatically extract or construct complex features through hidden layers, which is significantly different from the pattern-based method. The accuracy of these three kinds of identity-recognition methods has been increasing in recent years, with the approximate trend shown in Figure 4. It is obvious that higher accuracy is often achieved by the deep learning-based method, and it is becoming widely adopted.

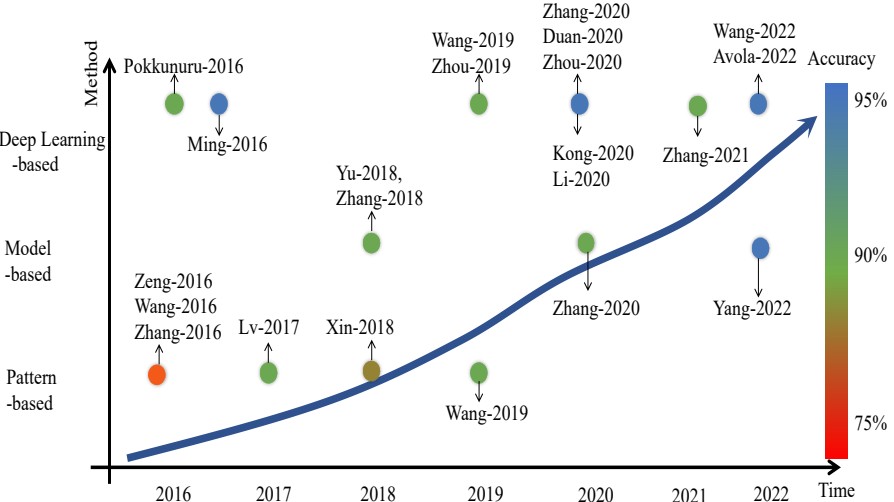

**Figure 4.** The trend of accuracy in a single-target scenario. Zeng-2016 [7], Wang-2016 [8], Zhang-2016 [9], Pokkunuru-2016 [13], Ming-2016 [15], Lv-2017 [11], Xin-2018 [12], Yu-2018 [17], Zhang-2018 [26], Wang-2019 [16], Wang-2019 [18], Zhou-2019 [19], Zhang-2020 [23], Zhang-2020 [20], Duan-2020 [21], Zhou-2020 [22], Kong-2020 [29], Li-2020 [29], Zhang-2021 [78], Yang-2022 [79], Wang-2022 [80], Avola-2022 [81].

In this paper, we survey state-of-the-art identity-recognition methods and categorize them into pattern-based methods, model-based methods, and deep learning-based methods. Note that despite the differences in their working principles and implementation methods, the state-of-the-art schemes in either pattern, model, or learning-based approaches can effectively capture gesture features and use these features for identity recognition; hence, they may exhibit overlapping performances in terms of accuracy (as can be noted from Tables 5–7). They nevertheless differ in other aspects. To begin with, the number of individuals in the dataset is different. Moreover, pattern-based methods usually need to establish a "hard-coded" pattern, which is predefined and often requires a lot of computational resources and time to optimize. Deep learning-based methods learn and optimize the model automatically through training data and automatically extract features from input data, so they have an advantage when dealing with large-scale, complex feature data. However, deep learning methods usually require much labeled data for training, with the risk of model overfitting. In addition, the transferability of these three methods is also different. For example, if a model is trained in a specific environment, it may not adapt well to a new environment. Deep learning methods, due to their self-learning ability, perform better in this regard.

### 4.1. Pattern-Based Method

4.1.1. Lesson Learned

The principle of pattern-based recognition is to find the patterns and the regularities in data. In most cases, this recognition method needs to assign a label to the given input data, and a recognition problem can then be viewed as a classification problem. For pattern-based identity recognition, tries to identify humans by exploiting CSI-changing patterns. Firstly, the data containing changes in CSI caused by human action is collected. Then, the regularity of CSI variation is determined and a unique mapping relationship between CSI variation and human action is established, based on which identity is recognized. Therefore, the key issue of this method is to associate the CSI signal change curve with a specific human action. How to describe different patterns to distinguish different actions is a challenge because the variation of CSI is complex. In general, the pattern-based identity-recognition method includes two processes: feature extraction and identity classification.

The existing works using pattern-based methods are summarized in Table 5.

**Table 5.** Summary of pattern-based models.

| Model (Year) | Signal | Pre-Process Method | Features (Network) | Number of T-R | Scale | Accuracy |
|---|---|---|---|---|---|---|
| WiWho [7] (2016) | CSI (amplitude) | Band-pass filter | Statistical features (Decision tree) | 3-3 | 2~6 | 92.0~80.0% |
| WifiU [8] (2016) | CSI (amplitude) | PCA, STFT | Walking speed, step length (SVM) | 2-3 | 50 | 79.28% |
| WiFi-ID [9] (2016) | CSI (amplitude) | Band-pass filter, CWT | Time-frequency features (SAC) | 3-3 | 2~6 | 93.0~77.0% |
| Wii [11] (2017) | CSI (amplitude) | Low pass filter, PCA, CWT | Information gain (SVM) | 3-3 | 2~8 | 98.7~90.9% |
| FreeSense [12] (2018) | CSI (amplitude & phase difference) | Low pass filter, PCA, DWT | Image segmentation (KNN) | 2-3 | 2~6 | 94.5~88.9% |
| WiPIN [16] (2019) | CSI (amplitude) | Butterworth filter, FFT, IFFT | Time-frequency features; (SVM) | 1-1 | 1~30 | 100~92% |

4.1.2. Current Methods

WiWho [7] uses Wi-Fi signals to sense human action and distinguish different human bodies by using CSI as a gait characteristics carrier, then achieves identity recognition. WiWho firstly uses multipath elimination and Butterworth band-pass filtering to denoise and obtain valid information containing reflective single-step and walk features, then employs a decision tree classifier to output identity predictions. The results showed that WiWho can identify individuals at an accuracy of 80% to 92% in groups of two to six people, and in most cases, a walking distance of only 2~3 m was enough to complete the identification.

WifiU [8] obtains CSI signals from commercial Wi-Fi devices, uses the PCA method to denoise, and then converts them into a time-frequency joint domain by using STFT, and further reduces the noise by spectral image enhancement technology. In the feature extraction stage, the starting point of walking, gait cycle time, and speed of trunk and legs are detected to extract gait features, and finally, SVM is used for classification. The experimental results show that in a 50 m$^2$ room, WifiU's accuracy is 79.28% in a gait dataset of 50 people.

Although WiFi-ID [9] also uses commercial Wi-Fi devices for gait recognition, it does not extract specific gait features but directly analyzes the entire walking action. In WiFi-ID, CSI data in the 20~80 Hz band is selected as an effective frequency range of personal gait features, and continuous wavelet transform and relief feature selection algorithms are used to extract the statistical characteristics of gait in different bands. Finally, an algorithm based on SAC is adopted to determine the user's identity. The results show that WiFi-ID's accuracy is 93% to 77% in groups of two to six people.

Wii [11] uses PCA and a low-pass filter for denoising, and then extracts multi-dimensional gait features from the time domain and frequency domain, respectively, and selects the most effective gait features according to information gain. This method uses extracted features to achieve stranger detection through GMM and identity recognition

through the SVM method based on an RBF kernel. Experimental results show that the Wii has an accuracy rate of 98.7% to 90.9% for single-target identity recognition in two- to eight-people groups.

Due to the user's movements in the Wi-Fi environment causing phase differences between received waveforms of different receiving antennas, FreeSense [12] analyzes them to realize human body detection. PCA, DWT, and DTW technologies are used to capture specific effects of human movements on surrounding Wi-Fi signals, which are analyzed to realize human recognition. In addition, FreeSense also proposes a line-of-sight waveform feature extraction model with human walking information. In FreeSense, experimental results show that human tests have an average false positive rate of 0.58% and an average false negative rate of 1.20%. As the group size changes from six to two, human identity-recognition accuracy increases from 88.9% to 94.5%.

WiPIN [16] is an operation-free identity-recognition method. Firstly, it pre-processes collected signals using Butterworth filter and multipath effect elimination (FFT, IFFT), and then extracts time-frequency domain feature extraction. Secondly, the SVM method is used to match the module for identity identification. Finally, the SoftMax function is employed to predict illegal users. The experimental results show that WiPIN can achieve a recognition accuracy of 92% on 30 users, with high robustness to various experimental settings and a low recognition time cost (less than 300 ms). The above five recognition algorithms have their advantages and disadvantages, respectively. Although WiWho [7] has high recognition accuracy, it requires the tester to move in a straight line in a specific area and cannot be used to track a user, because the recognition can be triggered only when the user enters or leaves the room. The innovation of WifiU [8] lies in converting the original Wi-Fi timing signal into a high-fidelity spectrum, which can then use the classic classification methods in the image field, but it does not perform well on a large group of data sets. WiFi-ID [9] uses the mute elimination algorithm to determine the length and starting point of the effective area for the original data, and the recognition accuracy is relatively high, but it does not consider the recognition scheme of the non-Los path, nor does it consider the larger recognition group and the robustness of recognition. The recognition accuracy of Wii [11] is further improved, and it has the function of identifying strangers, which makes the research of gait recognition based on Wi-Fi perception expand to the field of intrusion detection. Wipin [16] can classify authenticated users based on the learned threshold in advance, and reject illegal users who it has not seen before.

### 4.2. Model-Based Method

#### 4.2.1. Lesson Learned

The principle of the model-based identity-recognition method is to build a model that associates the signal space with the physical space, and then use this model to determine human identity based on the relationship between the received signals and human actions. Human action can be accurately identified by exploring the laws of physics based on physical or mathematical models. In state-of-the-art CSI-based action recognition studies, typical models include the Fresnel zone model, AoA, human breathing model, interaction model, CSI velocity model, and CSI activity model. The existing works using model-based methods are summarized in Table 6.

**Table 6.** Summary of model-based methods.

| Model (Year) | Signal | Pre-Process Method | Features (Network) | Number of T-R | Scale | Accuracy |
|---|---|---|---|---|---|---|
| WiID [17] (2018) | CSI (amplitude) | PCA, STFT | FFT (SVDE) | 3-3 | 5 | 92.8% |
| SiWi [26] (2018) | CSI (amplitude) | BW, PCA, DWT | Data Segmentation (SVM) | 2-3 | Legitimate users: Invaders: | 93.0%; 97.0% |
| WiDIGR [23] (2020) | CSI (amplitude) | Band-pass filter, PCA | STFT (SVM) | 1-3 | 3~6 | 92.83~78.28% |
| Wi-IP [79] (2022) | CSI (amplitude) | Savitzky Golay filter | Time domain features, SFS, SBS (SVM) | 1-3 | 2~6 | 100~82.5% |

4.2.2. Current Methods

WiID [17] is a user identity-recognition system by gestures based on Wi-Fi signals, and it can identify a user who is performing a predefined gesture. Firstly, PCA is used to pre-process raw CSI, and then speed time series features are extracted to distinguish the gestures of different users. Secondly, the Vector Distribution Estimation of RBF is used to generate a user classification model of each gesture. Finally, features of input gestures are evaluated to identify the user identity with the highest matching degree. The experimental results show that the average accuracy of WiID is 92.8% in four environments.

SiWi [26] is also an identity-recognition system based on Wi-Fi gesture perception. Firstly, raw CSI data are pre-processed by using the Butterworth filter, PCA, and DWT methods and then sliced. After that, HMM is used to recognize the actions of sliced data segments. Secondly, according to three basic activities (push hand, swing arm, and wave hand), the Fresnel model is employed to establish the association between individual gestures and identity labels, and then SVM is used for identity recognition. Experiments show that the average accuracy of SiWi is 93% and 97% for legitimate users and invaders, respectively.

WiDIGR [23] is an improved algorithm for gait recognition using Wi-Fi sensing technology. Firstly, band-pass filtering and PCA methods are used to pre-process raw data to reduce dimension and denoise. Secondly, STFT is used to transform 1D time series data into a 2D spectral map, and WiDIGR extracts gait feature information from an aggregated spectral map (both manually and automatically). Finally, SVM is employed for classification and identification. The results show that the accuracy of WiDIGR is 92.83% and 78.28%, respectively, in groups of three and six people. The advantage of WiDIGR is that it eliminates directional dependence, meaning that the gait can be identified no matter which direction testers are walking in.

Wi-IP [79] is an identity-recognition model based on CSI analysis of gait, which has achieved good results in the case of a small number of training samples. Its main processes include the original CSI data collection and pre-processing, amplitude features selection and extracting using SFS and SBS, and identity recognition using SVM with RBF kernel. The average recognition accuracy of Wi-IP in a group of approximately two to six people is from 100% to 82.5%.

Comparing the above six algorithms, we find that WiID and SiWi have similar approaches for identity recognition, both providing user identity classification models for each predefined gesture behavior. In addition, the CSI data segmentation mechanism proposed by SiWi can extract more accurate feature information, and the use of the Fresnel model to obtain the distance and direction of user behavior further helps achieve better action recognition results. However, WiID and SiWi still have limitations, such as large sample data volume requirements and limited types of recognized actions. Both WiDIGR and Wi-IP are improved algorithms for gait recognition using Wi-Fi sensing technology. The advantage of WiDIGR is that it eliminates directional dependence, enhancing the universality of the model. Wi-IP has achieved good results with a small number of training samples. However, both WiDIGR and Wi-IP still have the problem of insufficient utilization of feature data (only using amplitude information).

*4.3. Deep Learning-Based Method*

4.3.1. Lesson Learned

Deep learning is a type of machine learning algorithm that utilizes deep neural networks (such as autoencoders, CNNs, LSTMs, RBMs, etc.) to classify data. Typically, machine learning algorithms require accurate features as input because these features characterize the input data and determine the output, but deep learning typically does not require a feature extraction step as it can automatically discover and extract features from input data using neural network models. Deep learning enables a new classification method capable of handling large-scale data with complex features. The existing works using deep learning-based methods are summarized in Table 7.

**Table 7.** Summary of deep learning-based models.

| Model (Year) | Signal | Pre-Process Method | Features (Network) | Number of T-R | Scale | Accuracy |
|---|---|---|---|---|---|---|
| NeuralWave [13] (2016) | CSI (amplitude & phase) | PCA, Data Interpolation | Frequency-domain features (CNN) | 3-3 | 1~24 | 87.76 ± 2.14% |
| HumanFi [15] (2016) | CSI (amplitude & phase) | Butterworth filter | Frequency-domain features (LSTM) | 3-3 | 24 | 96% |
| CSIID [18] (2019) | CSI (amplitude) | Data sample selection, CSI time series conversion | Time series features (CNN, LSTM) | 1-3 | 2~6 | 97.4~94.8% |
| Deep-WiID [19] (2019) | CSI (amplitude) | Data sample selection, CSI time series conversion | Frequency-domain features (GRU) | 1-3 | 2~6 15: | 99.7~97.7%; 92.5% |
| Gate-ID [20] (2020) | CSI (amplitude & phase) | Low pass filter, linear transformation of phase | Spatial-temporal features (ResNet, BiLSTM) | 3-3 | 6~20 | 90.7~75.5% |
| WiNet [21] (2020) | CSI (amplitude) | None | Frequency energy diagram (CNN) | 1-3 | 40 | 98.5% |
| LWID [22] (2020) | CSI (amplitude) | None | Frequency energy diagram (CNN) | 1-3 | 50 | 98.8% |
| FingerPassf [29] (2020) | CSI (amplitude & phase) | low-pass Filter | IFFT (LSTM) | 1-1 | 7 | 90.6% |
| WiHF [30] (2020) | CSI (amplitude & phase) | Band-Pass Filter, PCA | STFT (CNN, GRU) | 1-3 | | 97.65% |
| WirelessID [78] (2021) | CSI (amplitude) | PCA, STFT | (CNN, LSTM) | 1-6 | 5 | 93.14% |
| Caution [80] (2022) | CSI (amplitude) | Dimension transformation | Few-shot Learning (CNN) | 1-3 | 2~15 | 98.34~86.29% |
| Re-ID [81] (2022) | CSI (amplitude & phase) | Data Sample selection | Amplitude heatmap, phase features (CNN, LSTM) | 2-3 | 35 | 90% |

### 4.3.2. Current Methods

According to the different basic network structures, the identity-recognition methods using the Wi-Fi signals-based deep learning method can mainly be classified into four types, i.e., CNN-based, LSTM-based, both CNN and LSTM-based, and others.

1. CNN-based method

These methods, often taking two-dimensional data as input, utilize basic CNN or its variants. Representative works include NeuralWave [13], WiNet [20], LWID [22], TransferSense [24], WiHF [30] and Caution [80].

NeuralWave [13] first pre-processes the collected signals (missing data interpolation, phase calibration, noise reduction), and then performs automatic feature extraction, including PCA dimension reduction and 1D deep convolutional neural network. Finally, the SoftMax classifier is adopted to realize user identity recognition. Experimental results show that NeuralWave can achieve a user identity-recognition accuracy of 87.76 ± 2.14%.

Considering that different subcarriers have different representations of gait features, WiNet [21] converts 1D time series CSI data into a 2D frequency energy map, to obtain richer features. The model is divided into two stages: frequency energy graph generation and gait recognition. In WiNet, the operations of convolution, regularization, activation, and global pooling are performed for the frequency energy graph, in turn, to effectively extract gait features. Last, a SoftMax function is employed to classify. The experimental results show that the accuracy of WiNet can reach up to 98.5% in a group of 40 people. In addition, WiN et al. so tests gait recognition in different scenarios and targets with additional items, and the average recognition rate is more than 92%, reflecting its robustness.

LWID [22] is a novel lightweight gait recognition model based on Wi-Fi sensing technology. Its data reconstruction method is similar to WiNet, which transforms original time series data into 2D images. LWID designs a bionic Ballon mechanism to cut a large number of neurons in the network layer and then combines convolution kernels of different sizes to integrate different features of channel information in feature ma. The experimental

results show that LWID can achieve a recognition accuracy of 98.8% in a group of 50 people, and the model size is only 6.14% of other recognition models.

TransferSense [24] is a gait recognition algorithm with strong generalization ability. Firstly, it performs amplitude filtering and phase calibration on raw data and takes the combined information of both as a feature. Secondly, CNN is used to extract features and realize gait recognition. Finally, transfer learning theory is employed for cross-scene perception. The experimental results show that the average accuracy of Transfer-Sense is more than 97% when the group size is 44. In other different environments, the average accuracy of target identity recognition is more than 77%. Compared with other similar algorithms, TransferSense not only performs well in the perception of a single target in a large group but also has a good generalization ability.

WiHF [30] is an algorithm for real-time cross-scene gesture and user recognition with Wi-Fi sensing technology. Firstly, raw CSI data are processed by band-pass filtering and PCA, and STFT is employed to analyze denoised data. Secondly, a method is proposed to capture features of gesture motion changes quickly and input them into a dual-task module for gesture recognition and user recognition. Finally, a recursive neural network is used to extract features and splice their respective features, and then the prediction results of recognition are output. The experimental results show that WiHF can achieve an accuracy of 97.65% and 96.74% in the same and different environments, respectively. It can be seen that this method has good recognition accuracy and generalization ability.

Caution [80] is a CSI-based human identity authentication system that uses only a small amount of gait CSI data. The system first collects data, then reduces the dimension of the data and maps the data to the feature plane to calculate the Euclidean distance, then compares the ratio with the intruder threshold, and finally classifies it by few-shot learning. The experimental results show that the accuracy of caution recognition is 86.29~98.34% when the population size is 2~15.

Comparing the above six algorithms, we find that: NeuralWave is the first in the literature to use deep learning for feature extraction and classification of physiological and behavioral gait biometrics embedded in CSI signals from commodity Wi-Fi. WiNet is the first to propose a data reconstruction strategy based on frequency energy maps, which enhances the feature description and accommodation capacity of sensing data while still maintaining a high recognition accuracy when the recognition scale is large. LWID, based on the WiNet data reconstruction strategy, focuses on the lightweight of the model; these three models all have the problem of insufficient utilization of feature data (only amplitude information). Both TransferSense and WiHF make full use of the amplitude and phase information of CSI data and have strong generalization capabilities in different scenarios. Caution uses a small amount of gait CSI data for classification through a few learning sessions. In the future, transfer learning and model light-weighting may become the focus of Wi-Fi identity perception research.

2.　LSTM-based Method

This type of method, which often takes the one-dimension data as input, is based on basic LSTM or its variants and mainly includes HumanFi [15], Deep-WiID [19], Gate-ID [20], FingerPass [29], article [28] and article [82].

HumanFi [15] is a new passive human body recognition method. Firstly, CSI measurements by commercial Wi-Fi devices are collected, amplitude denoising and phase calibration are performed in the pre-processing part, and a new gait detection algorithm based on a buffer and filter mechanism is proposed to solve the influence of short-term abnormal fluctuations. Finally, LSTM is adopted to distinguish time features of automatically extracted human gait features, and SoftMax is used to identify different people. Experimental results show that HumanFi achieved a 96% identity-recognition accuracy.

In the feature extraction stage, Deep-WiID [19] combines GRU and average pooling layer, which can automatically extract gait features and identify identities from CSI data and is more efficient than traditional manual feature extraction methods. The experimental results show that when the population size is from two to six, the average accuracy of Deep-

WiID is from 99.7% to 97.7%. When the number of people is 15, the average recognition accuracy can reach up to 92.5%.

Gate-ID [20] uses a theoretical communication model and practical measurement to prove that antenna array direction and walking direction contribute to the mirror mode signal in Wi-Fi. Firstly, the collected signals are pre-processed, including static elimination and segmentation, antenna array direction, walking direction analysis, and walking direction estimation. Secondly, an attention-based deep learning model is used to extract and enhance time-frequency domain features of the signal, then the SoftMax classifier is adopted to identify different people. The experimental results show that Gate-ID can uniquely identify people in a group of 6~20 people with an average accuracy of 90.7~75.7%, respectively.

FingerPass [29] is a model for identifying users based on gesture identification. Firstly, IFFT and Butterworth's filtering are applied to raw data. Secondly, the user recognition model is constructed by segmenting signals and the LSTM algorithm. Finally, SVDD is used to build a lightweight model for real-time user authentication. FingerPass has an accuracy rate of 90.6% in the same scenario and 87.6% in different scenarios. Overall, FingerPass makes the model lightweight, while maintaining high accuracy and good generalization.

In addition, Liu J et al. [82] proposed a method to extract human respiratory biological features from Wi-Fi signals, but that paper did not pay attention to extracting gesture features. Inspired by this, Liu et al. [28] proposed a method of identity recognition using Wi-Fi signals and gestures. The method uses three hand gestures in the game of Rock, Paper, Scissors to identify which player is making the gesture. Firstly, Butterworth low-pass filter and PCA are used to pre-process original data, then DTW and random forest are employed for feature extraction, and finally, identity identification is achieved by LSTM. The experimental results show that the accuracy of this method is more than 95% for 10 volunteers, and the average accuracy can reach up to 97.4%. The results indicate that, although gesture information is not as unique as gait information, it can also be used as an important feature for identity recognition under certain limited conditions.

Comparing the above five algorithms, we find that Both HumanFi and WiID use LSTM and SoftMax for feature extraction and classification. WiID is the first work to use deep neural networks for feature extraction and classification, which not only improves accuracy but also effectively reduces the workload of data pre-processing. CSIID, based on the short-cycle characteristics of behavior perception patterns, uses Long Short-Term Memory (LSTM), which effectively solves the problem of gradient explosion or disappearance when RNN models deal with long-term correlation issues. Deep-WiID and WiID have similar structures, but Deep-WiID has higher recognition accuracy and stronger robustness, which can be said to be an improvement on the WiID model. FingerPass achieves high-precision user recognition using a lightweight network. According to the research results in the literature [27], although the uniqueness of gesture information is not as good as gait information, under certain conditions, gestures can also be an important tool for identity recognition.

3. Both CNN- and LSTM-based Methods

This type of method typically takes one-dimensional or two-dimensional data as input, is based on CNN or LSTM, as well as their combined variants, and mainly includes Wirelessid [78], Re-ID [81], and CSIID [18].

Wirelessid [78] explores the human fine-grained action and physical characteristics embedded in the channel state information by extracting spatiotemporal features. In addition, it also introduces the attention mechanism. The system first collects the signal and then uses the denoising algorithm based on PCA to perform STFT to obtain DFS. The spectrum of DFS is used to extract spatiotemporal features, and then the attention space module and attention time module are used for identity recognition by CNN and LSTM. The experimental results show that the average accuracy is 93.14%, and the best accuracy of five people can reach up to 97.72%.

Re-ID [81] is a personnel re-identification system, which is composed of a two-branch connected structure and one expanded on this basis. Each model branch includes two paral-

lel subnetworks. The system first collects CSI data and extracts amplitude and phase, then anomaly detection and data difference processing are carried out to generate a heatmap representing a given person, which is analyzed through the CNN-based network. The experimental results show that the evaluation accuracy of the system is 90%.

CSIID [18] uses both a convolutional network and LSTM to automatically extract gait features from CSI data, and then the SoftMax function is adopted to classify the identity. The experimental results show CSIID identity-recognition accuracy is from 97.4% to 94.8% when the group size is from two to six.

Comparing the above three algorithms, we find that all three use CNN and LSTM neural network structures, but each has its strengths. Compared to CSIID, Wirelessid introduces an attention mechanism that makes the model more precise in selecting key features. Re-ID is the first work to demonstrate the direct use of Wi-Fi sensing technology for personal re-identification technology.

As we can see, the discrimination basis of the above methods is either target gait or target gesture. The differences between them are in data acquisition, feature extraction, and classification.

In terms of data acquisition, gait-based identity-recognition methods mainly repeat collecting walking data of different individuals, which belong to coarse-grained human action. Gesture-based identity-recognition methods collect data for a specific group of gestures, which are human action characteristics with medium granularity. Different gesture-based identity-recognition methods have different types and gesture numbers, which are different from the gait-based identity-recognition methods with a single movement type.

In terms of feature extraction and classification, gait-based identity-recognition methods mainly use both machine learning and deep learning, while gesture-based identity-recognition methods mainly use machine learning. The algorithms of feature extraction and classification using machine learning algorithms have low complexity and less training time, but their accuracy is not as good as that of deep learning. Although deep learning algorithms have higher accuracy, they take a long time to train.

## 5. Multi-Target Scenario

Although the CSI can provide more fine-grained sensing features than the RSS information, the potential performance is still greatly influenced by the multipath effects. The methods introduced in Section 4 can only identify a single target in the environment but not multiple targets simultaneously. When there are multiple targets in a Wi-Fi environment, the sensing data collected at the receiver not only includes multipath signals of the surrounding obstacles but also signals generated by mutual reflection between targets. When the distance between targets is small, the receiver may mistake multiple targets for a single- target, resulting in recognition failures. However, the multi-target sensing scenario is ubiquitous in practice. There are a few works on multi-target sensing using Wi-Fi signals, especially for identity recognition. Therefore, we introduce multi-target sensing tasks, including activity recognition and identity recognition, in this paper.

The main challenge for the multi-target sensing tasks is that the received signals are much more complicated than those in a single-target scenario. According to the processing ways of mixed signals, Wi-Fi-based multi-target sensing technologies can be categorized into direct recognition and separated recognition.

### 5.1. Direct Recognition

Direct recognition first extracts features directly from the received raw Wi-Fi signals and then digs out the activities of identities on these features. These technologies often recognize a fixed number of targets, limiting their practicality. The existing multi-target sensing techniques are mostly direct recognition, such as sleep monitoring [83–85], action recognition [10,86,87], indoor tracking [88], multi-target counting [89,90] and identity recognition [91], as shown in Table 8.

**Table 8.** Summary of multi-target recognition works.

| Model (Year) | Application | Equipment & Signals | Used Method | Number of T-R | Scale | Accuracy |
|---|---|---|---|---|---|---|
| TinySense [84] (2017) | Breathing detection | CSI (amplitude) | FFT, IFFT, TOA | 2-3 | 2 | 88.00% |
| Mais [10] (2017) | Action recognition | CSI (amplitude & Phase) | outlier filter, KNN | 1-1 | 3 | 93.12% |
| Literature [83] (2018) | Breathing detection | CSI (amplitude) | Hampel filter, FFT | 1-2 | 2 | error of 0.5~1 bpm |
| WiMU [87] (2018) | Action recognition | CSI (amplitude) | generating and combining virtual samples | 1-1 | 2~6 | 95.0~90.9% |
| MUFIC [86] (2019) | Action recognition | CSI (amplitude) | DWT | 2-3 | 1~5 | 93.0% |
| Literature [88] (2019) | Tracking | CSI (amplitude) | MUSIC, Particle Filter (PF) | 1-1 | 1~3 | error of 38 cm outdoors and 55 cm indoor |
| DeepCount [90] (2019) | Counting | CSI (amplitude and phase) | Butterworth filter, PCA, DWT, CNN-LSTM | 1-1 | 5 | 86.40% |
| Literature [85] (2020) | Breathing detection | CSI (amplitude) | subcarrier combination | 1-1 | 4 | 86.0% |

Wang et al. [84] proposed a multi-user respiratory detection system called TinySense, which uses CSI from multiple transmitters (TX)-receiver (RX) antenna pairs to obtain respiratory data of a target. TinySense can simultaneously detect the breath of two targets with more than 88% accuracy. However, these technologies require multiple TX-RX devices, which greatly limits large-scale implementation. Yang et al. [83] explored the relationship between Wi-Fi deployment and Fresnel zone location for sleep monitoring, with an average respiration rate detection error between 0.5 bpm and 1 bpm in a two-target case. Wang et al. [85] proposed a new system to continuously track multiple targets' respiratory rates. First, STFT is applied to extract periodic respiratory signals, generating a spectral graph. Secondly, the Markov Chain Model is introduced to deal with the dynamic problems in natural respiration. At the same time, a new IDP algorithm is used to track the respiratory rate of each target one by one. Finally, time domain information and a quasi-bilateral filter are employed to remove outliers and estimate the number and identity of targets. The experimental results show that the average accuracy of target counting is 87.14% and 86.58% in two different environments (campus laboratory and automobile), respectively. The identity-recognition accuracy is 85.78% for four targets on average.

MAIS [10] uses Wi-Fi signals to sense human actions (including running, walking, and hand movements) and can recognize multiple activities of different testers in the same environment. Three basic modules are included in MAIS: data processing, activity detection, and activity classification. The data processing module aims to smooth amplitude and calibrate the phase. The activity detection module is responsible for detecting activity starting location. The activity classification module sorts detected data using the KNN algorithm. The experiments show that MAIS can achieve an accuracy of 98.04% for anomaly detection, 97.21% for predicting people numbers, and 93.12% for predicting human activities. Wen et al. [86] proposed an online method for modeling the position-behavior feature of a multi-target scenario based on single CSI action features, with an average recognition accuracy of 93%. Venkat Narayan et al. [87] randomly combined single-user gestures to generate virtual samples of multi-target mixed gestures and then compared them with real samples for identity recognition.

There have been some studies on multi-target localization and tracking. For example, reference [88] is a method of multi-target tracking. Firstly, it constructs a two-dimensional

signal model using a transmitter and receiver array, and the model is used to estimate various AoA parameters (target position functions and motion direction). Then MUSIC algorithm is employed to estimate the above parameters. Finally, PF with JPDAF is used to track multiple testers walking in the experimental environment area. In multi-target recognition, target number determination is a precondition for identity recognition.

The literature [89,90] are studies of multi-target counting in a region. The literature [89] proposes WiCount, a method that uses CSI in Wi-Fi signals to identify the number of targets. In WiCount, the wavelet denoising method is used in the data pre-processing stage; amplitude fluctuation and signal distribution of CSI are analyzed in the signal analysis stage; mean, variance and range of CSI amplitude are extracted in the feature extraction stage; three classifiers (KNN, BP, SVM) are trained to recognize the number of targets in the classification stage. The literature [90] proposes a method named Deep-Count that uses Wi-Fi CSI signals to identify the number of targets. Deep-Count mainly includes a target recognition model and an error correction function.

Belal Korany et al. [91] propose a multidimensional framework, that can identify multiple targets through walls by a separated signal reflected from each target using off-the-shelf Wi-Fi devices, and achieve an average recognition accuracy of 82% in four different areas.

The direct recognition of mixed signals is greatly influenced by environmental factors, resulting in both unsatisfactory accuracy and limited scalability. These existing multi-target sensing studies, therefore, mostly focus on relatively primitive tasks such as localization, counting, and action recognition, where identification is not required. Accurate multi-target identity recognition remains a challenge.

### 5.2. Separated Recognition

Separated recognition identifies the activity of identity of a single target from the corresponding separated Wi-Fi signals. The existing works using the separated recognition method are very few due to the difficulty of the separation of the mixed signals and mainly focus on action recognition and respiratory rate detection.

Zhang et al. [92] used the Canonical Polyadic decomposition method to separate multi-target action signals and achieved a recognition accuracy of 88.25%. However, Wi-Run can only effectively decompose sinusoidal signals. Wang et al. [93] used CSI phase difference information between antenna pairs to generate CSI tensor data, from which the tensor decomposition was adopted to obtain the required respiratory signals to achieve high-precision monitoring of multi-user respiratory rate. Zeng et al. [94] deployed a pair of transceivers to address the problem of multi-target respiratory perception and separated the mixed signals through independent vector analysis to obtain respiratory rate information for each target. In a four-target scenario, MultiSense realized an average respiratory rate error of 0.73 bpm.

Overall, there are some existing studies, as shown above, but the research on multi-target task sensing is still in the early stage compared to the single-target sensing task and lacks a complete theoretical system. It may be a difficult problem to handle the multipath effect in a multi-target scenario for a long time.

## 6. Future Research Directions

Although there have been many studies of identity recognition based on Wi-Fi sensing, Wi-Fi technology is not designed in particular for identity sensing, causing some limitations for Wi-Fi identity sensing. In indoor environments, the effect of multipath wireless propagation is more complex, limiting identity-recognition accuracy. Accordingly, we highlight the research directions for future investigation in this section.

### 6.1. Transfer Learning

Since Wi-Fi sensing technology is greatly affected by the multipath effect. Most of the current sensing methods can achieve a high recognition accuracy after training in a

fixed indoor environment, but they perform poorly in other environments, limiting the application of the methods on a large scale. As shown in Figure 5, When a person performs the same action in different indoor environments, the obtained CSI amplitude significantly varies. Therefore, it will be interesting to explore the common hidden features of human gait or movement in different environments and establish effective sensing migration mechanisms to realize cross-environment transfer without training or with less training.

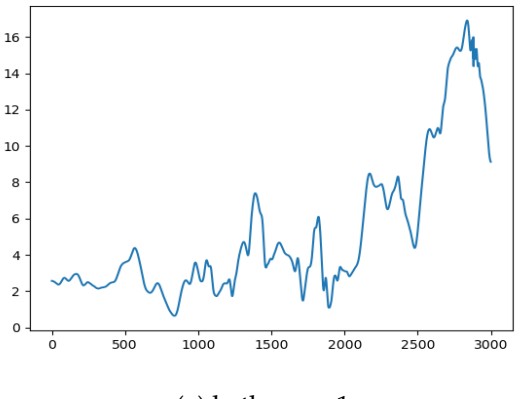
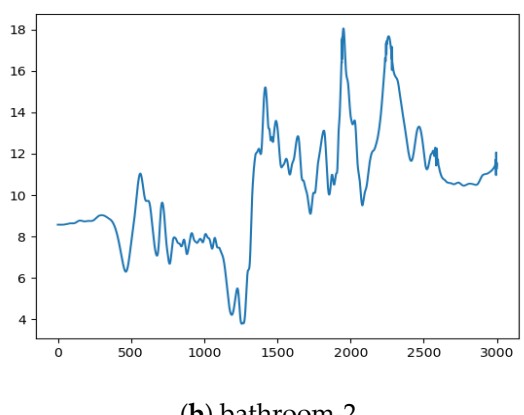

(**a**) bathroom-1                                                   (**b**) bathroom-2

**Figure 5.** CSI amplitude wave of the same person walking in different environments.

There have been several attempts for time series data-sensing model migration. In 2018, literature [95] attempted to migrate a deep learning-based algorithm for time series data and provided a beneficial experience of positive and negative migration. In 2019, the literature [96] elaborated migration experiments of various deep neural network algorithms on time series data sets in the form of a review and concluded that FCN, ResNet, and Encoder models have better migration performance for time series data. In addition, the literature [97] proposed a migration method based on network weights random initialization and conducted parameter combination experiments on six neural network models, i.e., ResNet, FCN, Encoder, MLP, time-CNN, and MCDCNN.

To realize effective migration of Wi-Fi sensing models in different environments, the fine-tuning methods that fixed neural network layers and parameters are often adopted in existing studies. In addition, environmental information effective removal, and accurate acquisition of human identity sensing features should also be studied in the future.

*6.2. Multi-Target Identity Recognition*

With the deepening of global aging, some more elderly people need to be taken care of. As a new contactless sensing technology, Wi-Fi sensing technology is suitable for meeting this need. For most aged-man families, there is often more than one person, meaning that the Wi-Fi sensing technology applied for multi-target scenarios is more suitable. Different from a single-target scenario, in which most of the noise is caused by the indoor environment's layout, the effect of mutual interference between targets will appear in a multi-target scenario, as shown in Figure 6. This effect will significantly increase the difficulty of identity recognition using Wi-Fi signals.

Therefore, to achieve more desirable multi-target recognition, it may be necessary to improve existing transceiver devices, which are commonly used for single-target recognition but not for multi-target. Considering that, it may be an improved direction, such as increasing the number of antenna arrays at the receiving end to improve the spatial resolution of target recognition and increasing the number of receiving ends to produce richer sensing information dimension, etc.

In a multi-target scenario, Wi-Fi signals are affected by not only the multipath effects but also the signals reflected among the targets. Therefore, the enhanced noise removal method is also needed in the future.

Most of the existing studies on multi-target are tracking, counting, and positioning, while there are relatively fewer efforts on human identity recognition. It is anticipated that multi-target recognition may receive more attention, due to its wider application range in practice than single-target sensing tasks.

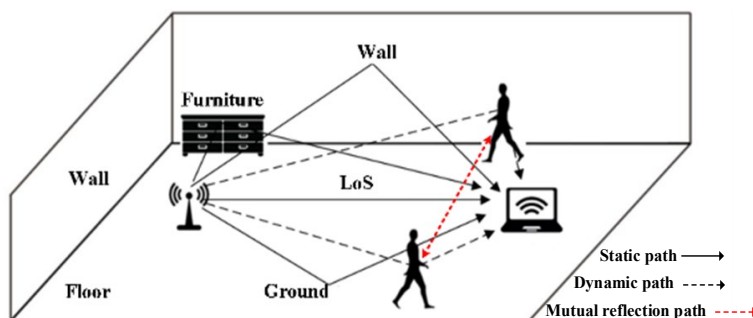

**Figure 6.** The wireless signals propagation path for a multi-target scenario.

### 6.3. Unified Dataset

A good dataset plays an important role in Wi-Fi sensing technology. An open, diversified, and accurate dataset guarantees convincing and reproducible experimental results. There are a few publicly accessible and unified datasets for validation of the Wi-Fi sensing models in current research, such as WiAR [98], WiDAR [99], etc. However, due to the multipath effects and the different experimental environments, these datasets cannot be well applied to other certain scenarios. Therefore, there is a strong need for more open datasets in the Wi-Fi sensing field, including ones for single-target scenarios and multi-target scenarios, respectively.

### 7. Conclusions

Human identity recognition is an important barrier to information security. An efficient identity-recognition approach has always been the pursuit of convenience. With the wide coverage of Wi-Fi signals, Wi-Fi sensing technology has become a natural research direction. With the advantages of low cost, contactless, free from light, and privacy-preserving, Wi-Fi sensing technology has been rapidly developing for human identity recognition. Most of these state-of-the-art efforts are based on gait and gesture. According to the number of targets, Wi-Fi sensing approaches can be classified into single-target identity recognition and multi-target identity recognition.

In this paper, we perform a comprehensive review of Wi-Fi sensing for human identity recognition. Firstly, the advantages and limitations of Wi-Fi sensing technology are discussed by comparing the existing identity-recognition technologies. Secondly, the main process of identity recognition in Wi-Fi sensing technology is elaborated. Thirdly, the current existing human identity-recognition methods for single-target scenarios and multi-target scenarios are introduced and analyzed in detail. In the future, more research may be conducted on aspects including effective removal of environmental information, accurate acquisition of human identity sensing features, and so on, to achieve effective migration of Wi-Fi sensing models in different scenarios. In addition, to achieve ideal multi-person recognition, the number of antennas at the receiving end can be increased to improve the spatial resolution of target recognition, which might generate richer dimensions of sensing information.

**Author Contributions:** Conceptualization, P.D. and X.D.; methodology, Y.C.; software, X.D.; validation, B.Z., J.K. and D.Z.; formal analysis, D.Z.; investigation, P.D.; resources, J.K.; data curation, X.D.; writing—original draft preparation, P.D.; writing—review and editing, B.Z.; visualization, D.Z.; supervision, J.K.; project administration, J.K.; funding acquisition, Y.C. All authors have read and agreed to the published version of the manuscript.

**Funding:** This research was funded by "Zhengzhou Collaborative Innovation Major Funding", grant number 20XTZX06013; "China Engineering Science and Technology Development Strategy Henan Research Institute Strategic Consulting Research Project" grant number 2022HENYB03; "Natural Science Foundation of Henan Province" grant number 222300420295 and Henan Science and technology research project grant number 232102210050.

**Data Availability Statement:** No new data were created or analyzed in this study. Data sharing is not applicable to this article.

**Conflicts of Interest:** The authors declare no conflict of interest.

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
