# Peer review of "A Comprehensive Survey on Wi-Fi Sensing for Human Identity Recognition"

_electronics, doi:10.3390/electronics12234858_

Round 1

Reviewer 1 Report

Comments and Suggestions for Authors

No major concerns.

Minor concerns:

1. Line 8: remove hypen in "human"

2. Table 1: What is "Non-Line-of-Sight"? Does it mean short distance? What is "body flavor". Is it body odor?

3. Line 96: Does the work by Chen belong to "Computer Vision"?

4. Line 204: Change "stably" to "reliably".

5. Line 277: Change "convention" to "conventional"

6. Line 291: What are "human organism"? Body parts?

7. Line 357: What is "Wii["

8: Line 452: Reference error

9: Line 560: Reference error

10: Line 661: Literature[86-87] should be plural.

11: Line 705: Change "seriously" to "greatly"

Comments on the Quality of English Language

See comments to authors. No major concerns.

Reviewer 2 Report

Comments and Suggestions for Authors

Overall Comments

This study provides a comprehensive survey on Wi-Fi-based human identification, which is an interesting topic in the field of human sensing. However, the manuscript should be refined for publication. The authors should explicitly state the problem statement and explain why Wi-Fi-based human identification is the focus of the paper. As this is a review paper, all information, knowledge, and results should be supported by appropriate references. High-resolution images with detailed information are necessary. Each section should provide detailed information and insights from the literature review rather than a summary of previous studies. The authors have used many images published in other papers, in this case, the authors should indicate the permission for their use.

Major Comments

  • There are some typos and sentences that need to be refined.

  • For lines 27-28, it is unclear what “explicit interaction” and “implicit interaction” mean in the fields of human-centered computing and human-computer interaction.

  • It would be better to compare other human identification approaches based on radio frequency with Wi-Fi-based approaches in Table 1.  

  • For lines 294-296, if most research on human gesture has focused on gesture classification, why does this paper explain gesture recognition for human identification?

  • For lines 286-287, why most previous studies have focused on single-target scenarios?

  • For Figure 4, what are the ranges for low, medium, and high accuracies?

  • In Tables 5, 6, and 7, the results show similar accuracies for the pattern, model, and learning-based approaches.

  • The manuscript mentions “Gait or Gesture” for human identification, but the authors used different words, including action or behavior, which leads to confusion about how Wi-Fi-based approaches have performed human identification

  • Figure 5 shows highly fluctuating CSI data, and it is crucial to clarify what activity this data represents. Is it human activity data?

Comments on the Quality of English Language

  • There are some typos and sentences that need to be refined.

Reviewer 3 Report

Comments and Suggestions for Authors

The authors raise a very important issue related to the use of WIFI networks for user recognition. The authors present a review, not a research article, so their work should be looked at a little differently.
In the Introduction, the authors sufficiently present the reason for the research and why the topic is important.
They go on to present the main methods of identifying people.
It is good that the work itself presents the identification process using WIFI.
In the subsequent parts of the article, the authors refer to appropriate sources.
The known methods are described in more detail below.
An interesting and appropriate description supported by appropriate results and comments.
The article ends with conclusions. This is where the authors could have come up with more developed conclusions.
In general, the article is interesting and suitable for publication as a review.
